# Mental health challenges and resilience strategies of Indigenous youth citizen scientists living in rural areas during COVID-19 school closures

Prasanna Kannan[1], Jasmin Bhawra[2], Kristi Wright[3], Tarun Reddy Katapally[4]*

**1** Johnson Shoyama Graduate School of Public Policy, University of Regina, Regina, Canada, **2** CHANGE Research Lab, School of Occupational and Public Health, Toronto Metropolitan University, Toronto, Canada, **3** Department of Psychology, University of Regina, Regina, Canada, **4** DEPtH Lab, Faculty of Health Sciences, Western University, London, Ontario, Canada

* tkatapa@uwo.ca

## Abstract

The COVID-19 pandemic profoundly affected youth well-being worldwide, yet the specific impacts on Indigenous youth remain underexplored. This study examined how school closures and online learning influenced the active living and mental health of Indigenous youth. Secondary school students (aged 13–18 years) from rural and remote Indigenous communities in Saskatchewan, Canada, participated as youth citizen scientists to explore these impacts and co-develop resilience strategies. Data were collected using a qualitative citizen science approach through virtual engagement in January 2021 and a follow-up focus group at the end of the academic year. Thematic analysis identified four key challenges during school closures: physical inactivity, academic difficulties, social isolation, and disrupted routines with notable gendered differences. Follow-up discussions revealed positive changes, including improved academic performance, increased family support, and the protective influence of cultural practices. The findings underscore the importance of culturally grounded, community-driven strategies to foster resilience among Indigenous youth. Practical implications include the need for inclusive school-based supports, recognition of cultural resources in education, and mental health policies that reflect youth voices. Future research should expand participatory approaches to better inform policy and programming tailored to the unique needs of Indigenous youth during and beyond public health crises.

## 1. Background

The COVID-19 pandemic presented an unparalleled challenge for global public health, with a profound impact on the mental well-being of young people [1]. The

**Data availability statement:** The study is part of the Smart Platform, a citizen science and mHealth initiative for ethical surveillance, integrated knowledge translation, and policy and real-time interventions. As this study contains sensitive data such as time-stamped location of citizen scientists, data requests should be sent to the University of Regina's Research Ethics Board at research.ethics@uregina.ca.

**Funding:** This study is funded by the Canadian Institutes of Health Research (CIHR) Project Grant #153226, the Saskatchewan Health Research Foundation (SHRF) Patient Oriented Research Leader Award, and the Canada Research Chair in Digital Health for Equity awarded to TRK. The funders had no role in study design, data collection and analysis, decision to publish, or preparation of the manuscript.

**Competing interests:** The authors have declared that no competing interests exist.

fears and anxieties brought about by the pandemic, coupled with school closure measures, led to a range of psychosocial and mental health issues in children and youth [2–5]. This vulnerable population experienced psychological distress characterized by fear and isolation, exacerbating stress, and hyperactivity due to school closures [6–8].

During these school closures, decreased physical activity among youth was also reported [9,10], which could have lasting implications for non-communicable disease risk [11–13]. Moreover, the pandemic exacerbated existing mental health disorders among children and youth, including anxiety disorders, depressive and stress-related disorders [1,14]. Notably, these mental health challenges disproportionately affected marginalized groups, particularly Indigenous communities [15,16].In the context of this study, "Indigenous peoples," a term encompassing the original peoples of North America and their descendants, have faced unique challenges during this crisis [17,18].

National data highlight persistent mental health disparities between Indigenous and non-Indigenous populations in Canada, rooted in intergenerational trauma, systemic discrimination, and social inequities [15]. During COVID-19, 38% of Indigenous participants reported poor mental health versus 23% of non-Indigenous counterparts, and 41% experienced moderate to severe anxiety [19]. These findings underscore the disproportionate mental health burden on Indigenous youth and the need for culturally grounded supports [18].

While these challenges are well documented, it is equally important to examine how children and youth have coped with the mental health impacts of the pandemic. General coping mechanisms include engaging in physical activity, maintaining social connections, practicing mindfulness, and turning to family or spiritual supports [20]. These strategies vary based on age, environment, and cultural background, and their effectiveness can differ across populations.

Indigenous youth, in particular, may draw upon culturally grounded coping strategies rooted in community, land-based practices, and intergenerational knowledge. However, research examining the specific coping strategies used by Indigenous children and youth especially those living in rural and remote communities remains limited. Generalized studies do not adequately capture the unique stress experiences and coping mechanisms shaped by culture [21,22].

The scarcity of specialized mental health services during the pandemic [23], combined with limited access to quality information and insufficient community input, has exacerbated mental health challenges like depression and suicidal ideation [15]. Cultural nuances deeply affect the mental well-being of Indigenous communities, underscoring the need for specialized psychological support, especially in emergencies [24]. Establishing universal protocols for psychosocial care during unprecedented situations like the COVID-19 pandemic is recommended [15]. Digital technology, including various digital tools such as mobile apps, offers promising avenues to strengthen mental health support for both the general youth population and specifically for Indigenous youth [25–27]. These technologies can provide improved access to mental health services while also preserving cultural ties [28]. This is

particularly beneficial for Indigenous youth, as it ensure that mental health support is culturally sensitive and tailored to their unique needs [29]. Notably, a strong sense of community belonging has been linked to improved mental health outcomes during the pandemic [30]. Indigenous individuals with a strong community connection exhibited lower anxiety symptoms, while those with a diminished sense of belonging faced a higher risk of depression [30]. Therefore, fostering a sense of community, especially in urban settings, is essential to support the mental health of Indigenous communities effectively [30].

Community-engaged research is crucial in empowering citizens, particularly marginalized groups like Indigenous youth, to ensure their representation in decision-making and inform policies that better support their mental health [29,31]. The term 'Community engagement' is defined as "the process of meaningfully involving communities affected by a research finding in the research process"[32].

Empowering Indigenous children and youth to have a voice in pandemic responses is imperative. Prioritizing their diverse perspectives aligns with the principles of Indigenous-informed restorative justice and ensures more robust and tailored solutions [33,34]. Moreover, encouraging youth involvement in matters that affect their lives is crucial in addressing the complexities of the current crisis [31,33,35]. Recent research highlights the importance of these young voices, as students' insights play a pivotal role in shaping effective strategies for school health and well-being [36,37].

This study captured Indigenous perspectives as part of a Smart Platform [38], an integrated platform for ethical community engagement, policy development, and real-time intervention [39]. Citizen science, which involves citizens throughout the research process, encompasses data collection, collaboration, and the co-creation of knowledge [40,41]. This transformative approach can help address societal crises, such as the COVID-19 pandemic, in disadvantaged rural and remote communities and amplify the voices of Indigenous youth to build resilience [42,43].

Digital citizen science has emerged as a powerful tool for ethically gathering large-scale data, complementing traditional scientific methods and enabling swift responses to complex global health dilemmas [44]. Given the pandemic's significant impact on education [45], there is a growing emphasis on citizen science tools tailored to assist educators and students in navigating the challenges of social isolation [46].

This study aimed to involve secondary school students from rural and remote Indigenous communities in Saskatchewan, Canada, as youth citizen scientists (YCS). The objectives were to 1) explore the challenges students face during school closures and their transition to online learning due to the COVID-19 pandemic for one academic year and 2) develop evidence-based strategies to enhance mental health resilience during the pandemic, using the amplified voices of youth to guide policy interventions addressing their challenges. This paper focuses on adapting the "Smart Indigenous Youth" (SIY) study [47], which is part of a Smart Platform initiative [38], seeking insights from Indigenous secondary school students about the implications of school closures, their health and well-being, the importance of their cultural contexts, and recommendations for school-based strategies to strengthen their resilience during the pandemic.

## 2. Methods

### 2.1. Ethics statement

Ethics approval was obtained for this study from the University of Regina and the Saskatchewan Research Ethics Board (REB#2017–29) [48]. This study was designed in collaboration with community partners, and its conduct reflects the spirit and letter of the Tri-Council Policy Statement 2 (TCPS 2) Chapter 9, which emphasizes respectful, reciprocal, and culturally sensitive engagement with Indigenous Peoples [49]. We also adhered to the Canadian Institute of Health Research (CIHR) guidelines for Health Research involving Indigenous Peoples [50], and the principles of Ownership, Control, Access, and Possession (OCAP) from the First Nations Information Governance Centre [51].

In alignment with TCPS2 Articles 9.12 to 9.16 [52], this research prioritized collaboration, mutual benefit, respect for Elders and knowledge keepers, and confidentiality. From the project's inception, elders were actively engaged, guided

community protocol, ceremonies, and knowledge translation. Community members including Elders, youth, educators, and leadership co-developed the research design, identified priorities, and shaped all study phases. The Tri-Council Policy Statement on ethical research with Indigenous communities outlines several important articles for conducting culturally-appropriate research. Written informed consent was obtained from the parent or guardian of each participant under 18 years of age who participated in the focus group of this study.

### 2.2. Overview of the Smart Indigenous Youth initiative

The Smart Indigenous Youth (SIY) initiative (2019–2023) is a 5-year community driven digital citizen science project co-created with Indigenous communities in Saskatchewan. Unlike traditional top-down frameworks, SIY was grounded in community-identified priorities to address youth mental health, land-based education, and cultural continuity. Its goal to integrate culturally responsive land-based active living programs into school curricula in rural and remote First Nations communities [48]. Its primary objectives are to promote mental health, reduce substance misuse, and prevent suicidal ideation among Indigenous youth. As a key component of the broader Smart Platform [38], SIY applies digital citizen science and mobile health technologies for ethical citizen engagement, real-time interventions, and holistic knowledge translation [39].

This project focuses on schools in First Nations reserves [53], considering the complex history of colonization that has increased the risk of physical and mental health problems among Indigenous people in Canada [54]. Central to SIY's methodology is the Two-Eyed Seeing approach [55] the initiative integrates Indigenous knowledge and Western methodologies, positioning Elders, educators, and youth as knowledge holders and co-researchers [56].

Extensive community engagement over three years informed the study's conceptualization, design, and data collection [39]. Community members are actively involved at every stage of research to ensure that the research process benefits participating communities [48].

Before the pandemic, trust-based relationships were established through repeated visits, ceremonial exchanges (ex: gifting tobacco), and community consultations. Decisions were made collaboratively regarding study objectives, design, focus group questions, and knowledge translation strategies. Youth and educators formed an advisory council that co-led methodological choices and reviewed results. This collaborative spirit extended to data ownership, ensuring that Indigenous communities retained control over their data [48].

The SIY initiative has been implemented in four schools since 2019, engaging students aged 13–18 in land-based active living programs. The Youth Citizen Scientist Advisory Council, comprising students and educators from all four schools, plays a crucial role in guiding the project. Feedback was gathered through focus group discussions and smartphone-based surveys. This approach ensured that all participants, especially youth and educators, were active contributors to the research process [48].

The Coronavirus disease (COVID-19) pandemic necessitated a pause in the land-based curriculum; however, the established relationships and infrastructure allowed the SIY project to adapt to a qualitative sub-study [37]. This adaptation enabled to capture of the school administration's response to COVID-19 and the subsequent decision-making processes to support youth mental health and active living. In this sub-study, youth citizen scientists were asked to share their challenges, resilience, and future actions they would like to take with the school to overcome barriers to mental health. The design and methodological overview of this study is outlined in Fig 1.

### 2.3. Design

This SIY sub-study used a natural experimental design [57–59], co-developed with school leadership and youth, to explore the impacts of school closures and online learning during COVID-19. This design was chosen to ethically examine the effects of the pandemic on Indigenous youth without manipulating exposure [60].

The COVID-19 pandemic led to school closure between March and September 2020. Following this period, participating schools reopened in October 2020 but shifted to online remote learning for the subsequent academic year. Due

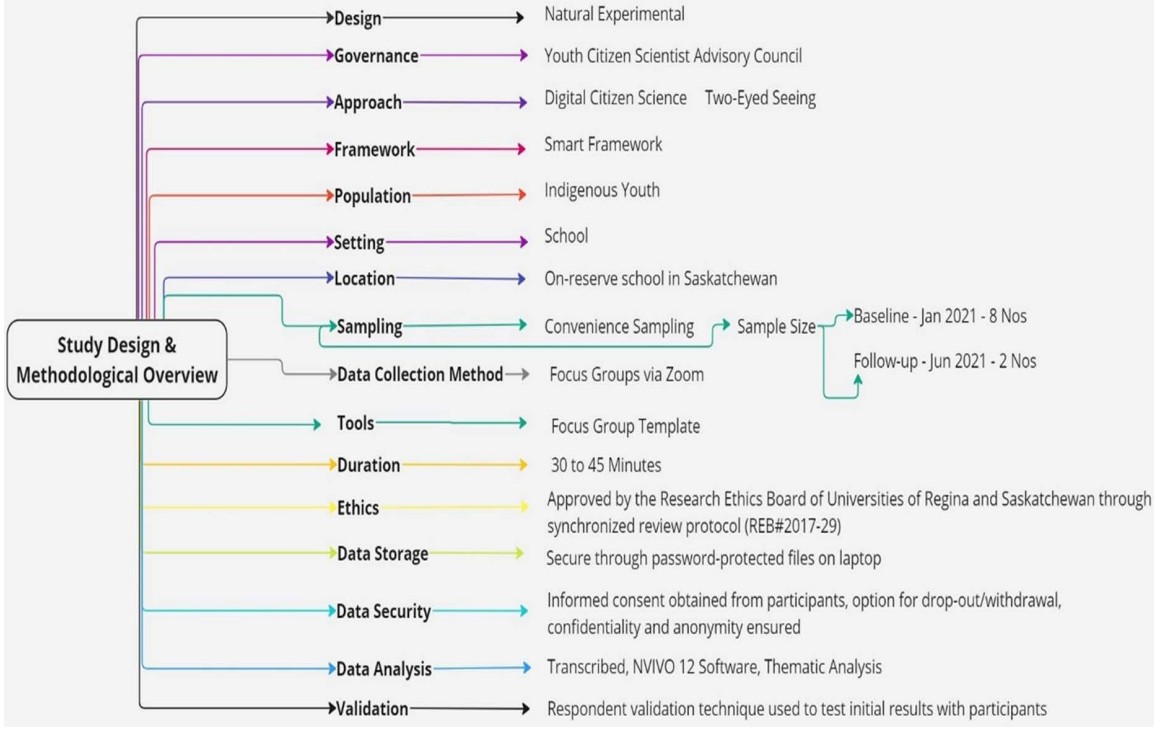

**Fig 1. Study design and methodology overview of Smart Indigenous Youth.**

to public health restrictions, external researchers were unable to enter communities. However, the established trust and digital infrastructure from the broader SIY initiative allowed the study to proceed virtually.

In January 2021, eight Youth Citizen Scientists participated in qualitative focus groups to offer retrospective insights into the difficulties posed by school closures and subsequent shift to online learning in the 2020–2021 academic year. This was followed by a focus group session in June 2021 with two YCS, who delved into the specific issues arising from the transition to online learning.

## 2.4. Study setting

SIY was conducted in schools located in Saskatchewan, Canada. This sub-study uses data from one of the Saulteaux/ Cree First Nations schools located 65 km North East of Regina, Saskatchewan and 15km west of Fort Qu'Appelle, Saskatchewan. The reserve currently has an area of 8,960 hectares, 2000 band members, approximately 69% of the membership live off reserve and is part of Treaty 4 territory. The school employs 19 teachers, 10 educational assistants and a variety of support staff to help approximately 175 students on their educational journey. Moreover, during the COVID-19 pandemic, the school adopted online learning as the mode of instruction for the academic year 2020–2021.

## 2.5. Theoretical framework

The SIY initiative is informed by the Smart Framework (Fig 2) [61].

The Smart Framework integrates citizen science, community-based participatory research, and systems science to enable data collection, stakeholder engagement, and integrated knowledge translation. A cycle of contribution, collaboration, and co-creation between researchers, citizens, communities, and policy makers is central to meaningful stakeholder engagement. Researchers play an important role in both citizen collaboration and knowledge translation back to

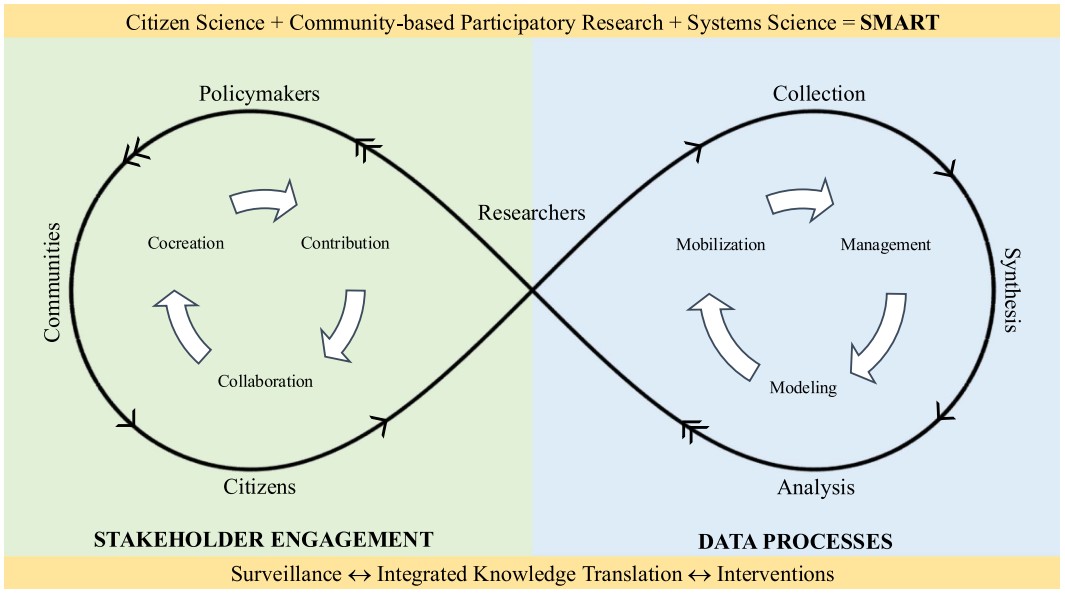

**Fig 2. The Smart Framework: integration of citizen science, community-based participatory research, and systems science via ubiquitous tools [61].**

all stakeholders. Together, stakeholder engagement and data processes facilitate collaborative development of research questions, data collection, and integrated knowledge translation via smartphones [61].

With respect to SIY, ubiquitous digital tools (e.g., smartphones) drive real-time engagement of citizen scientists in rural and remote areas, thus serving as tools of equity. The Smart Framework informs the co-creation of knowledge, where youth and educator citizen scientists residing in rural and remote areas partner with researchers. More importantly, in an effort to decolonize research methods in this study, using the Smart Framework, we have integrated citizen science with Traditional Indigenous Knowledge to ensure Two-Eyed Seeing for participatory action research [48,62]. Decolonizing research methods requires unlearning the Western conventions of data collection, analysis, and knowledge sharing to dissociate research from its colonial European roots [63].

Indigenous ceremony has been central to the design and delivery of the SIY project [64]. For instance, Elders, educators, and youth were engaged before conceptualization of the study, and played a key role in developing the engagement and data methodology [65,66]. Elders opened YCS Advisory Council meetings, and the team co-created knowledge in partnership with knowledge users (i.e., principals, educators, and youth), which has implications for informing and influencing culturally relevant policies. A decolonized lens to qualitative research was taken using focus groups to ensure equitable participation of all relevant Indigenous community stakeholders, and using a virtual talking circle format to encourage open conversation and unstructured dialogue [65].

## 2.6. Qualitative data collection

Purposive sampling was used to invite youth (Grades 9–12) who had previously participated in SIY and were recommended by educators and Elders for their interest and engagement. Eight YCS (two male, six female) joined the baseline focus group, and two YCS (one male, one female) joined the follow-up. Reduced participation in follow-up was due to academic scheduling conflicts and infrastructure challenges in remote areas such as unreliable Internet access.

The YCS Advisory Council, educators, and Elders co-designed focus group questions to reflect culturally grounded concerns and local lived realities. Sessions followed a virtual talking circle format, honoring Indigenous listening protocols,

reciprocity, and safe sharing. Each session was co-facilitated by a trained Indigenous educator and a research team member to ensure cultural safety and balanced power dynamics.

Participants gave informed consent electronically through an app; for minors, consent was also obtained from parents/guardians. Focus groups were conducted via Zoom [67], recorded with permission, anonymized, transcribed by the researcher (PK), and stored securely. The theme validation process included inter-coder consensus and review and interpretation by the YCS Advisory Council to ensure authenticity and cultural resonance.

This research reflects self-determination and ethical participatory engagement by embedding decision-making, data interpretation, and knowledge co-creation within community structures.

## 2.7. Data analysis

Thematic analysis process was conducted to systematically identifying, organize key themes from the focus group discussion following Braun & Clarke's guide [68,69]. Researchers generated a coding manual based on key themes which emerged from data analysis. Using this manual, two researchers then fully coded the transcripts to verify to identify themes and sub-themes. Reviewers met online to resolve any discrepancies, ensuring a consensus on the final themes and sub-themes. The data analysis was facilitated using the NVIVO version 12 software [70]. Analyses for each focus group were conducted separately and then combined to develop overarching themes and sub-themes. To ensure the accuracy and consistency of the themes, a third researcher reviewed the thematic coding and the identified themes and sub-themes.

Moreover, the YCS Advisory Council reviewed the results to enhance that the shortlisted themes were accurately captured [71,72]. Saturation of themes was considered achieved under two primary conditions: a) the recurrence of specific themes across the data indicated by frequency counts, where themes like lack of motivation in academic and active living perspectives repeatedly emerged until no additional prevalence was noted (e.g., sub-theme 1 (n = 4), sub-theme 2 (n = 2)); and b) when consensus or corroborative evidence from the literature supported the relevance and sufficiency of specific themes despite limited mentions (n = 1), such as 'digital literacy and competency' retained under the 'school closure academic challenges' sub-theme based on established literature recognition [73]. This dual approach ensured that no further interviews or data gathering provided new themes or insights, indicating saturation was achieved [74–76].

## 3. Results

The YCS participated in the focus groups in January 2021 in two groups (males and females) and retrospectively discussed the challenges faced during the school closures during the onset of COVID-19. Six themes emerged from the focus groups (Fig 3): *school closure challenges, health during COVID-19, support, future concerns, positive amid COVID-19, and coping strategies.*

In the subsequent follow-up in June 2021, the YCS discussed challenges related to online learning and strategies adopted to mitigate these challenges. Three themes emerged from these focus groups: *well-being, academic performance, and challenges, and online vs. in-person learning.*

The following is a detailed presentation of the findings from baseline and follow-up engagements.

### 3.1. Baseline focus groups with the YCS

**3.1.1. Theme-1 – school closure challenges.** YCS faced distinct challenges during the initial stages of the COVID-19 lockdown due to the school closures. This overarching theme branched into two subthemes: Barriers to active living and academic difficulties (Fig 4).

During the COVID-19 school closure, the challenges faced by youth citizen scientists varied by gender (Table 1). However, some of the concerns are similar for both genders. The males highlighted their struggles with physical inactivity,

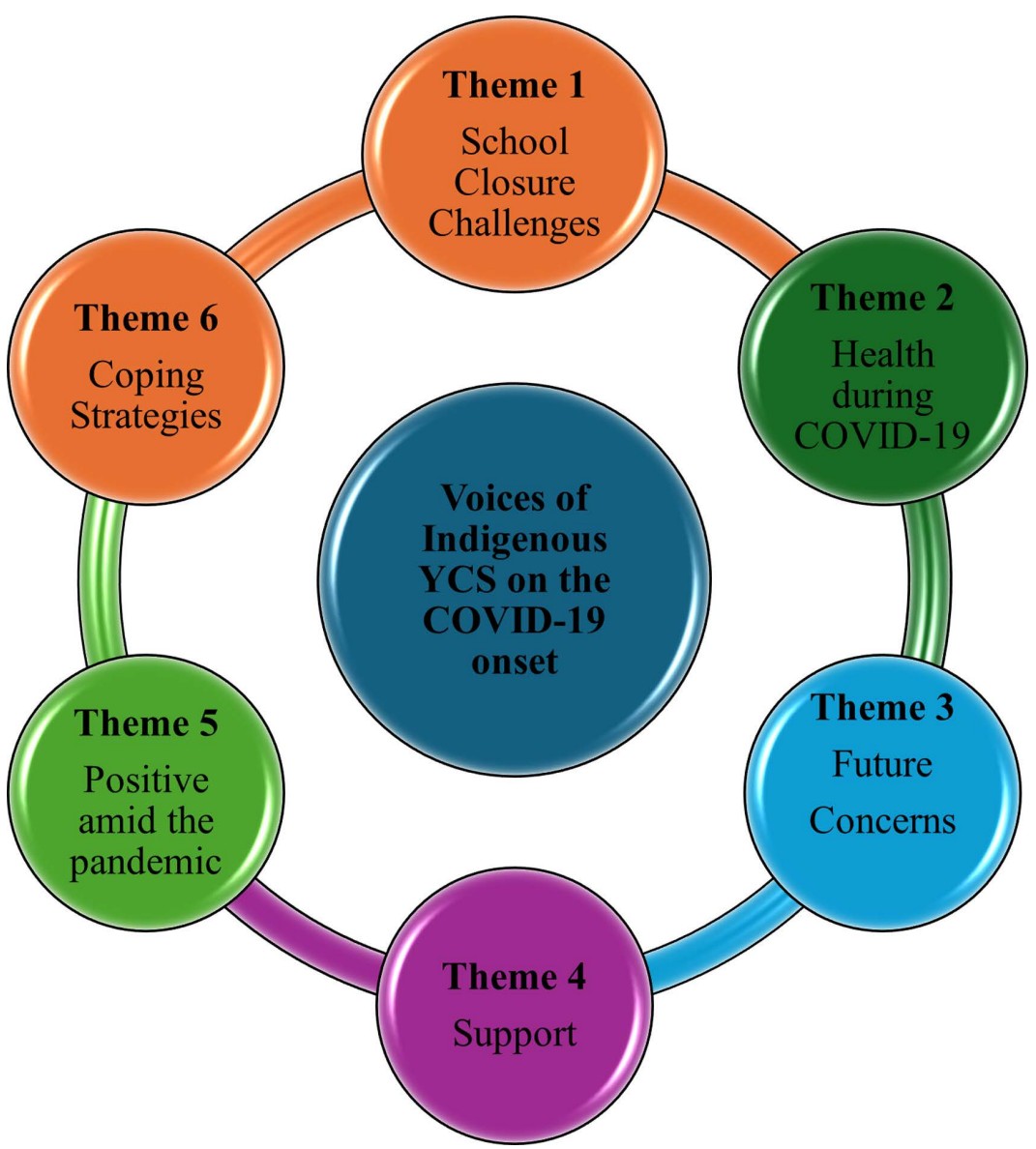

**Fig 3. Themes of voices of Indigenous Youth Citizen Scientist on the COVID-19 'onset'.**

expressing concerns about being unable to play sports or engage in other physical activities. YCS 2 noted that: *"COVID has affected my life in ways where I, I had to stop playing all my sports and being physically active, like, I used to play club volleyball. And then, like, I used to play a lot of volleyball around our communities we have here, and I've just stopped doing that, which affected me lots and stopped being physically active for a long time and haven't really gone anywhere much, which is very boring." (YCS 2)*

Subsequent prolonged inactivity made them feel restless and bored at home. In addition, they face academic challenges. One YCS Stated: *"For me, it's like, I find it a lot more difficult than being in class. Because I have to do like, where I do my work, it's in my room. And like my bed in my game and stuff like that are all right there." (YCS 2).*

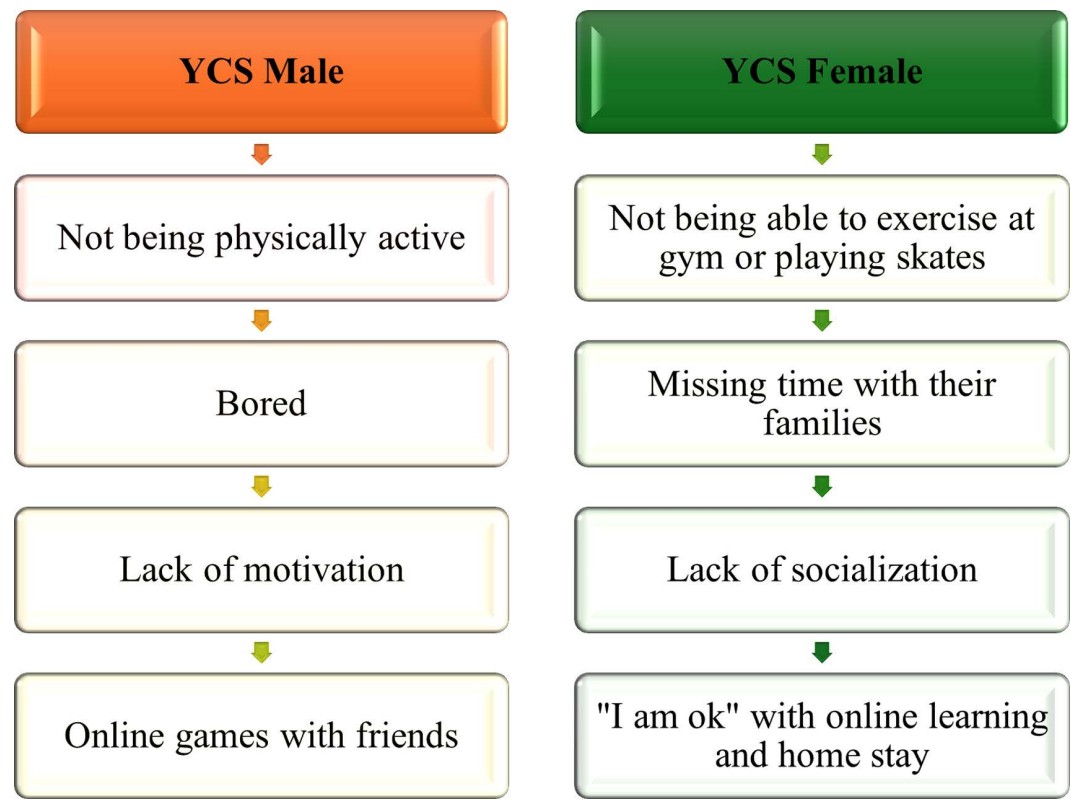

**Fig 4. School Closure challenges for YCS Male Vs Female.**

**Table 1. Sub-themes on academic challenges and active living barriers of Youth Citizen Scientist on the COVID-19 school closure.**

| Sub-theme 1 Academic Challenges | | Sub-theme 2 Active living barriers | |
|---|---|---|---|
| YCS Males | YCS Females | YCS Males | YCS Females |
| Lack of motivation | | Social distancing barriers for playing games | Not being able to exercise at the gym/outdoors |
| Internet access difficulties | Struggle and frustration in doing online work | Physical health challenges | Lockdown rules (staying inside the house) |
| Hard to focus on studies | Lower grades | Lack of motivation | |

Furthermore, YCS 2 added: *"And like it's hard to find the motivation to do though to do my work when I just want to like lay down or play a game with my friends and it's just a lot more difficult than actually having to go to school" (YCS 2).*

On the other hand, female citizen scientists' primary concerns revolved around the lack of social interactions and disruptions in their routines. They missed spending time with their families and attending school in person. The absence of social settings at school, inability to meet friends, lack of physical exercise, and unavailability of fitness facilities deeply affected them.

However, some of the concerns are similar for both genders. YCS expressed contentment while staying at home and adapting to online classes. YCS 5 felt that: *"I feel like it helped me in a way it helps me realize like what was real towards*

*what was fake you know, like things like that I blocked out like unnecessary people in my life, and I feel like it did the best for me"* (YCS 5).

**3.1.2.  Theme-2 - Health during COVID-19.**  At the onset of the COVID-19 outbreak, YCS males shared their perceptions of health. Initially, they felt healthy; however, prolonged inactivity during the school closure made them feel less healthy (Fig 5).

One YCS shared that: *"I sleep all day long and I have trouble breathing, it bothers me because it makes life harder, my mental health is no very good"* (YCS 8).

Another YCS felt that: *"I think COVID affecting my health because I don't get that much exercise i'm kind of lay in bed a lot because yeah don't really get to go places, because you have to stay home can't go visiting people either.um. I guess you could say affected my mental health, too, because you're not really interacting with people much. Just kind of staying in your room in…"(YCS 6)*

For YCS females face individualized challenges. Predominant among these were a sense of isolation from friends and family, an inability to access medical consultations, and decreased physical activity. Extended periods of inactivity and the absence of support manifested a noticeable deterioration in their mental well-being (Fig 5).

One YCS shared that: *"I can't like go to the doctors for a certain thing and I also can't like buy things I need for myself"* (YCS 3).

**3.1.3.  Theme-3 -Future concerns.**  The YCS males primarily expressed concerns about missing out on graduation ceremonies, particularly the traditional ones. The prospect of missing this significant rite of passage resulted in feelings of stress. They were also apprehensive about the unpredictability of COVID-19 (Fig 6).

Conversely, the main anxieties for YCS females revolved around apprehensions about COVID-19 vaccines, fueled by misinformation suggesting lethal outcomes. They also expressed concerns about returning to school, ongoing uncertainties about the pandemic, and potential future unemployment. However, some females were more focused on the present, setting aside concerns about the future (Fig 6).

One YCS shared that: *"I don't know the vaccine the COVID-19 vaccine is like kind of scary for me. Because like even though that I'm not old enough to take it it's still scary because say if a family member took it I'd be scared for them, because I heard, but like it killed, some people. So, like yeah and plus like COVID-19 comes down a bit we can get back to normal"* (YCS 3).

Another YCS felt that: *"Not being able to like to get a job or struggling. In the future, with something that you have to deal with now and basically, just about your mental health of how you feel now and, in the future,"* (YCS 4).

Addition to that, one YCS felt that: *"Like I'm not going to be stressing myself over what's going to happen in the future, when you could just live in the present you know"* (YCS 5).

**3.1.4.  Theme-4 - Support.**  Theme 4, Support, comprises two sub-categories: (a) support from family and friends, and (b) cultural support, each described in detail below.

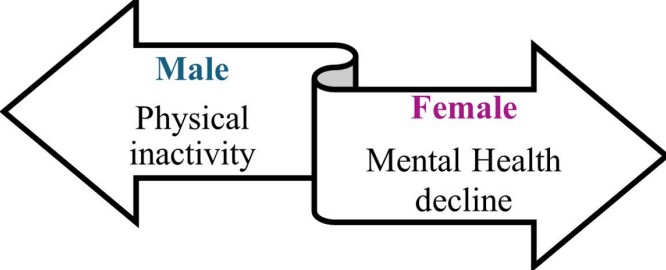

**Fig 5.  YCS Male Vs Female main concern affecting health during the COVID-19 pandemic.**

***Support from family and friends***: During the challenging times of COVID-19, the Youth Citizen Scientists' tendency to seek help varied. While males generally refrained from seeking assistance, females turned to multiple sources for support. They most frequently leaned on family, friends, cousins, and partners.

One YCS noted that: *"I want to support to my sisters, I have four, so it wasn't that hard, because obviously it's the same age so yeah I don't really find it hard" (YCS 5).*

Another YCS share that: *"My grandma surely and my sister she's my cousin, but I take her as a sister because I'm really close with her and then my other cousin that lives with her yeah that's about it" (YCS 4).*

YCS shared that: *"Some things that make me happy is I live with my brother and he has a son, my nephew is turning three soon. And like just playing with him. He's like always running around and yelling and stuff like that. Playing with him. Makes me happy. And sometimes we I blast music loud and he just dances real hard and makes me happy. And I like to like to play games online with my friends, because we always have fun time. We're always laughing around with each other. And yeah, just makes me feel better" (YCS 2).*

However, it is worth noting that some females felt that they did not receive the support they needed and often isolated themselves.

YCS 6 added that: *I really need to go to anybody for support, but like both of my parents work too so they've gone most that day, but like I'd go to my partner, ….and my best friend …she's the best person to be around" (YCS 6).*

Another YCS noted that: *"uhhh...no one except myself. I keep to myself" (YCS 8).*

Furthermore, YCS have shed light on community-driven efforts to encourage active living amidst the pandemic. Following public health guidelines, many community members maintained a sense of normalcy. This involved initiatives such as clearing ice from beaches to create makeshift skating rinks and setting up goal rings. Youth groups and band members also contributed by delivering packages filled with art and craft materials to homes, encouraging residents to stay active. Despite these efforts, some YCS observed limited activities within their communities.

***Role of culture***: Culture serves as a vital support system during challenging times such as the COVID-19 pandemic [77]. For instance, the YCS males found solace at cultural gatherings and although these events became less frequent and limited to smaller groups, they gave participants peace and connections (Fig 7). Nevertheless, the infrequency made the youth yearn for more community interaction.

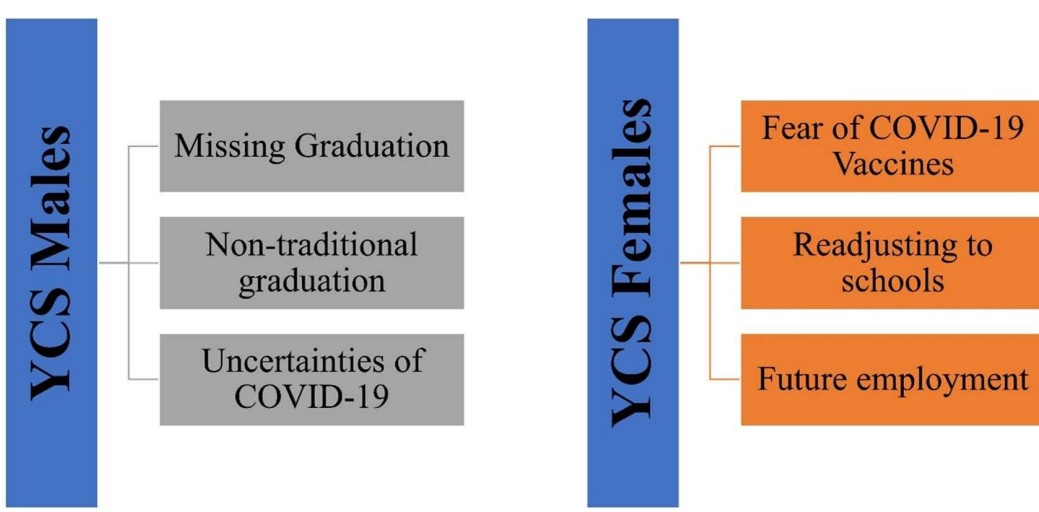

**Fig 6. Future concerns of the COVID-19 pandemic.**

One YCS felt that: *"Because I used to go to those a lot. Now. I just barely don't because when I used to go on July I feel at peace and now there's barely of it…"* (YCS 1)

Furthermore, smudging was identified as a key cultural practice during the pandemic, and the YCS felt that smudging removed negative energy and helped them feel better. Engaging in smudging not only uplifted the spirit of the youth but also strengthened their familial bonds.

YCS 7 shared that: *"Um well my family or my mother, my mother likes to smudge a lot and she does it to bring like to take away bad like negative energy she does that a lot and I guess you can say it helps. Yeah, that's like one of the but that has played a big role in like my family smudging. It helps a lot"* (YCS 7).

YCS 4 added that: *"My dad smudges the house time to time and to get rid of those like negative thoughts spirits are like stuff like that and I smudge time to time, but often yeah"* (YCS 4).

Likewise, the YCS females highlighted the significance of cultural support during these trying times. Community leaders played a pivotal role in ensuring that individuals received the necessary supplies, including food and other essential materials (Fig 7).

**3.1.5. Theme-5 - Positives amid the pandemic.** Despite the challenges of the pandemic, the YCS females noted some silver linings. YCS 3 felt that: *"I don't know I think it's kind of easier like this"* (YCS 3).

They reported enhanced academic performance and heightened concentration in their studies, feeling less burdened by everyday stressors. YCS 5 expressed that: *"I like doing online I don't really have a problem with it my grades are pretty high so like I like it because I get to go on my own pace, you know things like that so yeah I like online work"* (YCS 5). Another YCS perceived that: *"…people learn differently too so. But I kind of like this, because you don't have to go out it's the same thing as school"* (YCS 6).

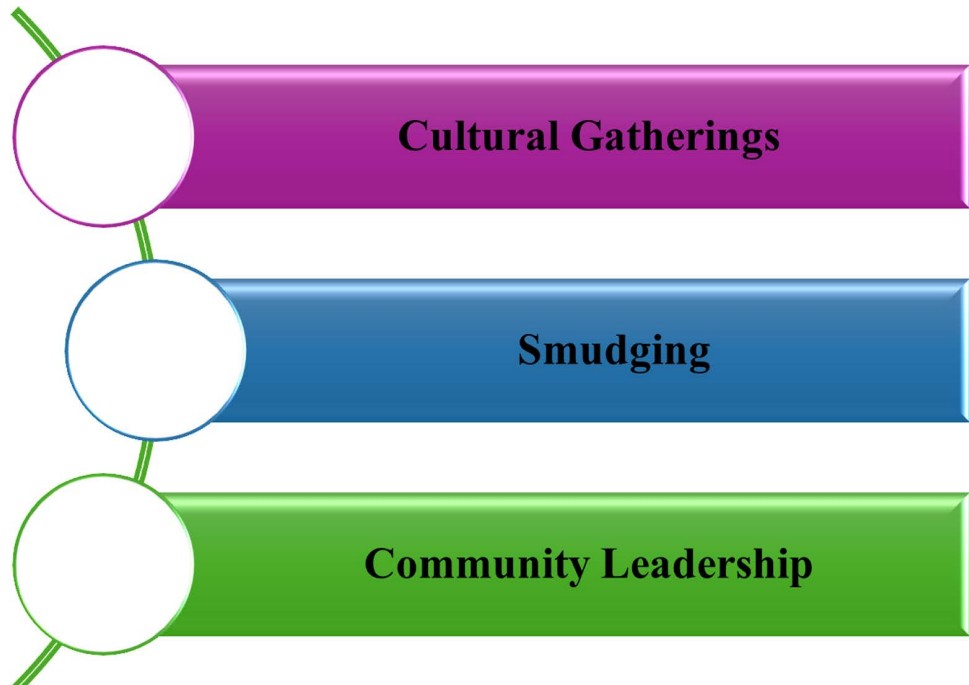

**Fig 7. Role of culture during the COVID-19 pandemic.**

Additionally, they distanced themselves from individuals who had a detrimental impact on their mental states. Collectively, both male and female YCS believed that the pandemic granted them an opportunity to introspect and deepen their connections with friends and family.

**3.1.6. Theme 6-coping strategies.** Youth Citizen Scientists have highlighted various strategies used to navigate the challenges of the COVID-19 pandemic (Fig 8). Family plays a pivotal role in YCS males. The pandemic allowed them an extended period to bond with their families and strengthen their relationships. Engaging in playful activities with siblings and joining friends for online games served as effective coping mechanisms.

YCS 6 share that: *"Maybe, just like going outside and being outside and then outside yeah going, and I feel like that really helps me just to get some fresh air and just to clear my mind and not be inside"* (YCS 6).

YCS 6 also felt that: *"Keep active would be going around. I started working out like two weeks ago. Honestly, that it makes me feel really good, because after I'm done working out. I feel more awaken more energized and just makes me feel good about myself each day becoming more better yep"* (YCS 6).

YCS 4 shared that: *"My mental health has like gotten a lot worse and I also watch anime to keep me distracted from like feeling negative thoughts and I kind of do my makeup time to time, but not all the time because there's hard times, or I have getting out of bed having to get that motivation and I basically keep healthy happy by volleyball in my sports because volleyball keeps me distracted and I get to see my friends and I get to like be happy basically yeah"* (YCS 4).

On the other hand, YCS females turned to the outdoors for solace, taking walks with family members.

They also immersed themselves in hobbies, such as drawing, painting, watching anime, and listening to music. Acts of self-care and the comforting presence of family were essential to their well-being. YCS 3 shared that: *"I guess like I'm drawing makes me happy. I like. Sometimes I paint. That like makes me happy enough, I guess, in person like sometimes. Not all the time, but like I go outside"* (YCS 3).

YCS males and females emphasized that participating in cultural ceremonies with their families was a therapeutic practice, boosting their mental health during these challenging times.

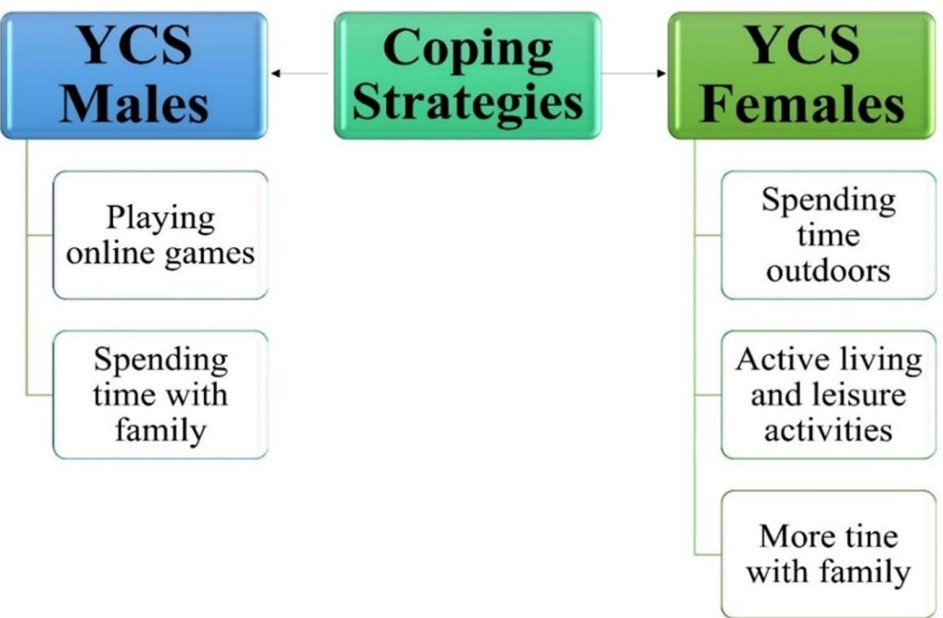

**Fig 8. Coping strategies used by Indigenous Youth Citizen Scientist during the COVID-19 pandemic.**

### 3.2. Follow-up focus groups with the YCS

In a follow-up focus group at the end of the 2020–2021 academic year, we engaged with two YCS to capture potential changes in the impact of online learning on their adaptability and resilience. Fig 9 below shows the three themes that emerged from in-depth conversations with the YCS to understand their valuable perspectives.

**3.2.1. Theme 1-well-being.** Youth Citizen Scientists shared their feelings about their well-being and coping during the pandemic. In addition to spending quality time with their loved ones, they have also been able to concentrate on their schoolwork. Furthermore, their parents encouraged them to be physically active, eat nutritious foods, and lead an active lifestyle. YCS 4 expressed that: *"…my mom has been making us walk the track and like basically running me and my brother and eating better as well. And I've been getting, you say more nutrients, or I don't play the game as much anymore as well" (YCS 4).*

Sporting and playing games with new friends brought happiness to the YCS. Notably, they felt increased support from both teachers and parents throughout the COVID-19 crisis.

YCS 4 expressed that: *"…… I would just go to my teacher or my mom because they're basically like my supporters. And then when I would go see my mom, she'll calm me down and explain to me like, well, I would have to get done and get it done" (YCS 4).*

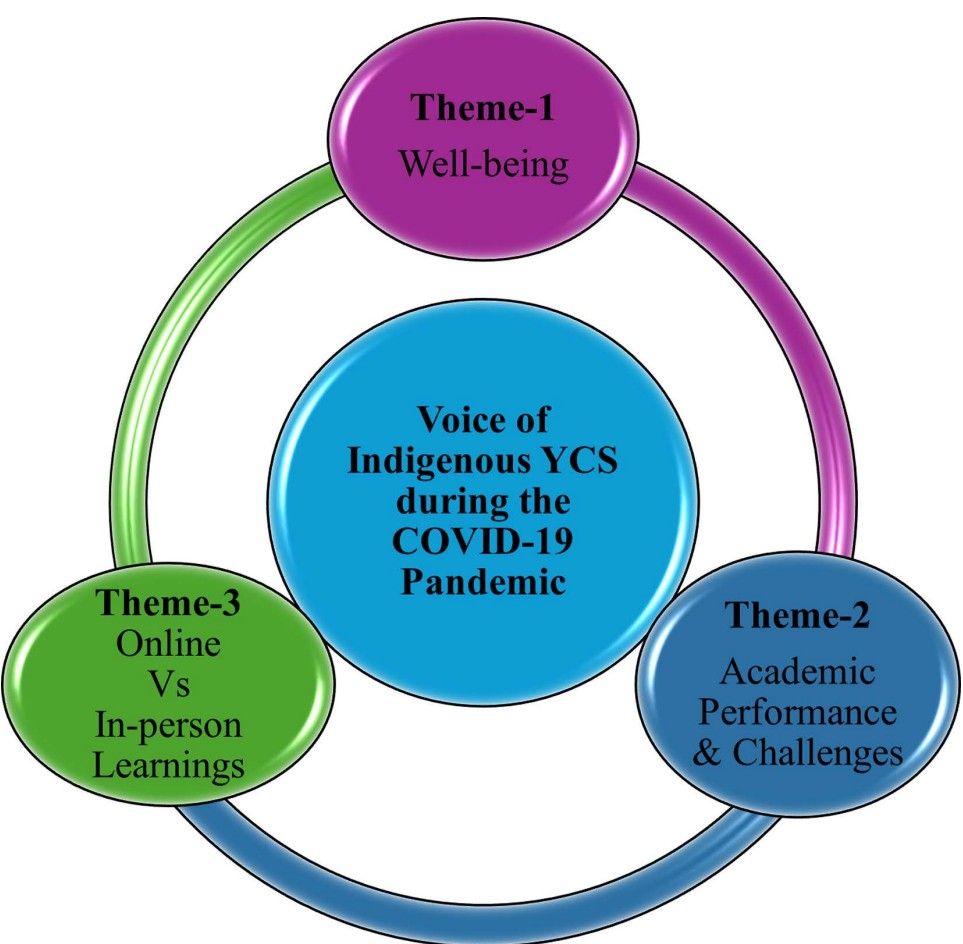

**Fig 9. Themes of voices of Indigenous Youth Citizen Scientist during the COVID-19 pandemic.**

### 3.2.2. Theme 2-academic performance and challenges.

Online education has enhanced the academic achievements of youth citizen scientists. The virtual learning environment has enabled many students to improve their grades and concentrate more on their studies. Additionally, they benefited from increased individualized attention from their teachers during online sessions. Some students noted improvements in completing assignments due to the flexibility and support provided in the virtual setting. YCS 4 expressed appreciation for the personalized support: *"My school years been pretty okay. But I've been having ups and downs. So I'm stressing out a couple times this past year. That's been pretty good. And my teacher has been helping me when she got on with me at 11 at night, and I was really thankful for her to help me with that. She basically the questions that needed to be answered in I was on my way to getting work done"* (YCS 4).

As a result, some YCSs managed to finish their school tasks ahead of schedule. However, for others, completing assignments from home presented challenges. YCS 1 noted that: *"Because we know home for me it's not really good. It's not really healthy. Because... when I'm at home, it should be my own time…"*(YCS 1).

### 3.2.3. Theme 3 - Online vs in-person learning.

Youth citizen scientists had mixed feelings about attending online classes and in-person learning at school. Some prefer to participate in online education as they are less likely to be distracted by their friends. While others believe that learning at home does not seem healthy and, thus, prefer to learn at school. YCS 1 expressed that: *"…While for me, I need to listen to see what task I need to do and what needs to be done. While I was at school, I didn't need to be distracted. Because the teachers are right there explaining what I need to be doing and what needs to be done. Even if we're visiting sometimes. It's all right, but you should be listening to the rest of the day. And also, you should be at school every day. Because we know home for me it's not really good. It's not really healthy. Because when I'm at home, it should be my own time. Right? It should be but since COVID, you know, came in? You guys are gonna do work at home…"*(YCS 1).

## 4. Discussion

This research highlights the profound impact of pandemics on the mental well-being of Indigenous youth emphasizing the need to explore effective strategies to promote positive mental health during pandemics, which aligns with other research findings [2,60,78–81]. The closure of schools during the pandemic disrupted formal education and raised concerns among youth about falling behind academically. The findings from the baseline focus group analyses illuminate the unique challenges faced by Indigenous youth during school closures caused by the COVID-19 pandemic. Similar to the present study, previous research has highlighted how school closures disproportionately affect vulnerable and marginalized populations, including Indigenous communities [78]. The disruption of daily routines, limited access to resources, and reduced social interactions have been identified as key challenges that impact academic progress and overall well-being [2,82].

The findings of this study suggest that males and females handle challenges differently and use distinct coping strategies [83–86]. Therefore, it is essential to develop age-appropriate and gender-specific interventions to help youth during challenging times such as the COVID-19 pandemic, aligning with other studies recommending tailored mental health strategies tailored for children and adolescents [79]. Considering the adverse effects of COVID-19 on children and youth's movement [87] and play behaviours, it is imperative to preserve and promote youth health during the pandemic and mitigate potential harm in future pandemics [88]. The findings of this study contribute to education policy and programs aimed at mitigating potential harm during future pandemics by improving movement behaviour [89,90]. Moreover, one of the study's key findings indicates the role of culture in creating resilience amongst youth to cope with pandemic challenges. Resilience [91,92] is defined as "the ability to positively cope, find hope, and foster constructive outcomes in contexts of adversity" [93]. Cultural practices, such as smudging was believed to remove negative energy and make YCS feel better [94]. Aside from improving mental health, this practice also helped youth connect with their families. Hence, culture acts can serve as a protective factor in improving mental health and well-being of YCS. The pandemic-induced closure of schools has also provided an opportunity for intergenerational learning as families reconnect and engage in reciprocal

relationships [95]. Previous studies have similarly found that engaging in culture during crises is associated with higher happiness levels. Therefore, cultural practices can promote mental health at the micro level, and social capital resilience at the macro level [16,96].

Digital engagement that aims to overcome technological literacy and competency constraints has also promoted the active living and mental health of YCS. Furthermore, it serves as a tool for developing coping mechanisms to overcome difficulties. These results align with studies suggesting that digital interventions can support population mental health during the COVID-19 pandemic [27,28,97]. Digital tool can address equity-related concerns, as digital exclusion, such as limited access to a computer or the internet, contributes to social isolation [98]. Smartphones, in particular, [98] offer opportunities to connect with youth during the pandemic and use digital citizen science to understand and mitigate challenges related to COVID-19 [43]. Similarly, the findings suggest that digital technologies such as smartphones, can effectively improve Indigenous youth mental health when applied with culturally-appropriate approaches [25]. The YCS recommend school-level measures dedicated to providing recreational facilities and physical activity opportunities to prepare for future pandemics such as COVID-19.

## 4.1. Policy implications

This study reinforces the importance of integrating Indigenous youth voices in policy and service design to improve health and well-being outcomes. In alignment with the "Desperately Waiting" report and Saskatchewan's "Children and Youth Strategy," establishing youth advisory councils and using digital citizen science platforms can foster meaningful participation in decisions that impact youth. Supporting youth voices through digital tools ensures more inclusive, culturally responsive governance and helps develop services that reflect the diverse needs of Indigenous youth. This also highlights the need for digital inclusivity. Governments must prioritize investment in digital infrastructure, literacy, and affordability to bridge the digital divide, especially in rural and remote Indigenous communities. Equitable access to technology is vital for supporting mental health, cultural continuity, education, and healthcare access, particularly during emergencies like the COVID-19 pandemic.

Policies must promote cultural integration within education systems by embedding Indigenous traditions, practices, and knowledge into online learning platforms. Culturally sensitive mental health services and community-based youth councils involving Elders, Traditional Knowledge Keepers, educators, and youth should be expanded across regions. Flexible learning models that accommodate hybrid options can empower students with more autonomy over their learning. Additionally, policies should support physical activity through culturally relevant programming and educator training to better serve Indigenous students. Ongoing feedback mechanisms from students and their families and sustained investment in research will ensure policy and education strategies remain adaptive, equitable, and grounded in the realities of Indigenous youth.

## 4.2. Strengths and limitations

This study involves Indigenous youth as citizen scientists in a natural experiment setting [69] during the COVID-19 pandemic, allowing for the co-evaluation of the impact of school policies and programs on promoting youth health and well-being. This ensures the inclusion of often-overlooked voices of Indigenous youth in pandemic discourse. Utilizing a comprehensive temporal analysis, this study identifies challenges at distinct phases of the pandemic. It emphasizes the urgent need for mental health interventions in schools by leveraging educational institutions as crucial platforms for supporting student mental health [70]. Employing tools such as Zoom [67], the study engaged with Indigenous youth virtually. This facilitates real-time understanding of challenges and enables informed, timely decision-making through integrated knowledge translation. Finally, by delving deeply into the personal narratives of the Indigenous Youth Citizen Scientists, the study provides a profound understanding of the complex challenges and resilience strategies during the pandemic. It addresses a significant gap in current literature, highlighting a historically marginalized demographic population.

Despite its strengths, the study also presents some limitations. The study's exclusive focus on students attending online classes restricts the scope, as it does not encompass the experiences of students in hybrid or in-person learning environments. Additionally, it overlooks the experiences of those returning to school during the academic year. Future research should expand to these areas to provide a more comprehensive understanding of diverse educational experiences during the pandemic. While small sample sizes are often viewed as a limitation in research, in the context of this study, they serve as a strength. The smaller size enabled deeper engagement with Indigenous youth as citizen scientists during the challenging COVID-19 pandemic. This approach facilitated rich data collection, which was particularly valuable given the difficulty in accessing many communities at this time. The success in engaging these youth is attributable to the strong relationships and partnerships within Indigenous communities, which enabled meaningful data collection and insights. However, the unstable Internet issues faced by rural communities profoundly affected our interactions with the YCS, resulting in fewer participants for follow-up.

Utilizing the Zoom platform for real-time engagement led to connectivity issues, particularly with participants from rural and remote locations. These internet connectivity and bandwidth constraints [71] hindered students from linking with the researcher and impacted their academic activities. Moreover, these connectivity challenges posed difficulties in scheduling follow-up sessions with participants.

## 5. Conclusions

This study, through its citizen science approach, has highlighted the experiences and coping mechanisms of Indigenous youth in Saskatchewan during the COVID-19 pandemic. By actively involving these youth as citizen scientists, the research gathered in-depth insights into their challenges during school closures and the shift to online learning, empowering them to contribute to knowledge creation. This participatory method underscores the importance of their voices in the research, ensuring that the findings are deeply rooted in their lived experiences. This study emphasizes the necessity of integrating cultural support and community engagement, alongside the utilization of digital tools, to bolster the mental health and resilience of Indigenous youth. It advocates policy frameworks that prioritize youth participation, digital inclusivity, and cultural practices, ensuring that interventions are tailored to the unique needs of Indigenous communities.

## Supporting information

**S1 Appendix. Focus group template.**
(DOCX)

## Acknowledgments

The principal knowledge users of the Smart Indigenous Youth initiative are the school principals, the File Hills Qu'Appelle Tribal Council, and the Saskatchewan Ministries of Health, Education, and Sport. The Youth Citizen Scientist Advisory Council is involved in implementing Smart Indigenous Youth, and in shaping school policies and programs. We are grateful to the youth, educator, and administrator citizen scientists, and the staff and trainees of the Digital Epidemiology and Population Health Laboratory (DEPtH Lab) for their continuous support.

## Author contributions

**Conceptualization:** Prasanna Kannan, Jasmin Bhawra, Tarun Reddy Katapally.

**Data curation:** Prasanna Kannan.

**Formal analysis:** Prasanna Kannan.

**Funding acquisition:** Tarun Reddy Katapally.

**Investigation:** Prasanna Kannan, Jasmin Bhawra.

**Methodology:** Prasanna Kannan, Jasmin Bhawra, Tarun Reddy Katapally.

**Project administration:** Tarun Reddy Katapally.

**Resources:** Prasanna Kannan, Jasmin Bhawra, Tarun Reddy Katapally.

**Software:** Prasanna Kannan.

**Supervision:** Tarun Reddy Katapally.

**Validation:** Prasanna Kannan, Jasmin Bhawra.

**Visualization:** Prasanna Kannan.

**Writing – original draft:** Prasanna Kannan.

**Writing – review & editing:** Jasmin Bhawra, Kristi Wright, Tarun Reddy Katapally.

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
