## [Decision Letter · Decision Letter 0]

18 Mar 2025

PMEN-D-24-00569

Mental Health Challenges and Resilience Strategies in Rural Areas during COVID-19 School Closures: Indigenous Youth Citizen Scientists Insights from a Natural Experiment

PLOS Mental Health

Dear Dr. Katapally,

Thank you for submitting your manuscript to PLOS Mental Health, We apologize for the delay in reaching a decision. After careful consideration of the reviewer reports, we feel that your paper has merit but does not fully meet PLOS Mental Health’s publication criteria as it currently stands. Therefore, we invite you to submit a revised version of the manuscript that addresses the points raised during the review process. Please ensure that all comments raised by the reviewers are addressed in full.

We look forward to receiving your revised manuscript.

Kind regards,

Karli Montague-Cardoso

Executive Editor

PLOS Mental Health

Journal Requirements:

 1. Please include a complete copy of PLOS’ questionnaire on inclusivity in global research in your revised manuscript. Our policy for research in this area aims to improve transparency in the reporting of research performed outside of researchers’ own country or community. The policy applies to researchers who have travelled to a different country to conduct research, research with Indigenous populations or their lands, and research on cultural artefacts. The questionnaire can also be requested at the journal’s discretion for any other submissions, even if these conditions are not met.  Please find more information on the policy and a link to download a blank copy of the questionnaire here: https://journals.plos.org/plosmentalhealth/s/best-practices-in-research-reporting. Please upload a completed version of your questionnaire as Supporting Information when you resubmit your manuscript.  2. Please amend your detailed Financial Disclosure statement. This is published with the article. It must therefore be completed in full sentences and contain the exact wording you wish to be published. i. Please clarify all sources of funding (financial or material support) for your study. List the grants (with grant number) or organizations (with url) that supported your study, including funding received from your institution. ii. State the initials, alongside each funding source, of each author to receive each grant.iii. State what role the funders took in the study. If the funders had no role in your study, please state: “The funders had no role in study design, data collection and analysis, decision to publish, or preparation of the manuscript.”iv. If any authors received a salary from any of your funders, please state which authors and which funders.  3. Please make sure the funding information on the submission form matches your financial disclosure statement. Please indicate by return the full and correct funding information for your study and confirm the order in which funding contributions should appear. Please be sure to indicate whether the funders played any role in the study design, data collection and analysis, decision to publish, or preparation of the manuscript.  4. Please insert an Ethics Statement at the beginning of your Methods section, under a subheading 'Ethics Statement'.  5. Please provide separate figure files in .tif or .eps format. For more information about figure files please see our guidelines:   https://journals.plos.org/mentalhealth/s/figures https://journals.plos.org/mentalhealth/s/figures#loc-file-requirements    6. Tables should not be uploaded as individual files. Please remove these files and include the Tables in your manuscript file as editable, cell-based objects. For more information about how to format tables, see our guidelines:  https://journals.plos.org/mentalhealth/s/tables  7. In the online submission form, you indicated that “The data supporting the findings of this study are available upon request from the corresponding author.”  All PLOS journals now require all data underlying the findings described in their manuscript to be freely available to other researchers, either 1. In a public repository, 2. Within the manuscript itself, or 3. Uploaded as supplementary information. This policy applies to all data except where public deposition would breach compliance with the protocol approved by your research ethics board. If your data cannot be made publicly available for ethical or legal reasons (e.g., public availability would compromise patient privacy), please explain your reasons by return email and your exemption request will be escalated to the editor for approval. Your exemption request will be handled independently and will not hold up the peer review process, but will need to be resolved should your manuscript be accepted for publication. One of the Editorial team will then be in touch if there are any issues.  8. Some material included in your submission may be copyrighted. According to PLOS’s copyright policy, authors who use figures or other material (e.g., graphics, clipart, maps) from another author or copyright holder must demonstrate or obtain permission to publish this material under the Creative Commons Attribution 4.0 International (CC BY 4.0) License used by PLOS journals. Please closely review the details of PLOS’s copyright requirements here: PLOS Licenses and Copyright. If you need to request permissions from a copyright holder, you may use PLOS's Copyright Content Permission form. Please respond directly to this email or email the journal office and provide any known details concerning your material's license terms and permissions required for reuse, even if you have not yet obtained copyright permissions or are unsure of your material's copyright compatibility.  Potential Copyright Issues: a. We do not publish any copyright or trademark symbols that usually accompany proprietary names, eg (R), (C), or TM (e.g. next to drug or reagent names). Therefore please remove all instances of trademark/copyright symbols throughout the text, including (“OCAP®”) on page 11.

Additional Editor Comments (if provided):

Reviewers' comments:

Reviewer's Responses to Questions

**Comments to the Author**

1. Does this manuscript meet PLOS Mental Health’s publication criteria ? Is the manuscript technically sound, and do the data support the conclusions? The manuscript must describe methodologically and ethically rigorous research with conclusions that are appropriately drawn based on the data presented.

Reviewer #1: Partly

Reviewer #2: Yes

2. Has the statistical analysis been performed appropriately and rigorously?

Reviewer #1: N/A

Reviewer #2: N/A

3. Have the authors made all data underlying the findings in their manuscript fully available (please refer to the Data Availability Statement at the start of the manuscript PDF file)?

Reviewer #1: Yes

Reviewer #2: No

4. Is the manuscript presented in an intelligible fashion and written in standard English?

Reviewer #1: Yes

Reviewer #2: Yes

5. Review Comments to the Author

Reviewer #1: PNEM-D-24-00569

Mental Health Challenges and Resilience Strategies in Rural Areas during COVID-19 School Closures: Indigenous Youth Citizen Scientists Insights from a Natural Experiment

Overall comment

This is an interesting article. It not only focuses on mental health challenges of COVID 19, but takes it a step further to investigate the resilience of learners through the pandemic. Even though authors did a good job, I suggest a few changes that would strengthen their work. The article would also benefit from a language editor.

Title

The title does reflect information found in the article. However, consider swapping a few words to make it simpler and easier to comprehend. eg Mental health challenges and resilience strategies of Indigenous Youth Science Citizens Scientists living in rural areas during COVID-19 school closures.

Abstract

Even though all elements required in an abstract are included, I would suggest that authors trim it down and only leave information that speaks to the following; aim of the article, one or two lines that show the importance of this aim, methods of data collection used (including demographics such as sample size, age range). Findings (themes), then briefly situate findings in literature (one or two lines), why your study is crucial given these findings, who will/can benefit from your research (optional). This would reduce the length and increase its effectiveness.

Background

I suggest you strengthen how you transition from discussing mental health challenges of COVID 19, to coping mechanisms and how you position your study. For instance, you could follow your discussion of challenges with a discussion of children and youth’s coping mechanisms (in general), then narrow it down to Indigenous youth and children. This clearly defined funnel approach would then position your study better.

Methodology

While authors do make reference to sampling and focus group discussions, there is a need for a more thorough description of their decisions related to sampling, and focus group discussions in 2.6.

Results

Here are a few copy and paste tweaks that would strengthen your findings.

Theme 1

Consider only using two quotes instead of 3 if they all advance a similar point. Reconsider YCS3‘s quote under ‘females missing social interaction’. It does not support that point. Also provide data that supports this next point or consider deleting it. “However, some of the concerns are similar for both genders”.

Theme 3

Consider placing the first two quotes immediately after “They most frequently leaned on family, friends, cousins, and partners”. Then move your sentence of “However, it is worth noting…” directly above YCS 6’s quote. Consider moving some quotes from theme 7 (coping mechanisms) to this theme because they also refer to support. For instance, playing and taking walks with family and friends was a facilitative to both males and females’ resilience. Then rethink your statement that males did not rely on family support.

Theme 4- consider swapping theme 4 with theme 3 so that you group risk themes together and resilience themes together.

Theme 5

Theme 5 is still a form of support. Rethink your layout. For instance- I would make support the main theme. Then have subcategories: a) family and friends support b) cultural support

Follow up FG

Theme 2- consider starting YCS4 quote from “and my teacher…”. Add a quote or 2 that support this statement…”some YCSs managed to finish their school tasks ahead of schedule”.

Discussion

Consider removing the first paragraph and start the section strong with paragraph 2.

Even though discussing policy implications is important, consider compressing this information into one or two paragraphs. Then write a separate article that delves deeper into policy implications.

Reviewer #2: It was a pleasure reviewing your work entitled: “Mental Health Challenges and Resilience Strategies in Rural Areas during CO School Closures: Indigenous Youth Citizen Scientists Insights from a Natural Experiment,” a study that explores the perspective of Indigenous youths’ mental health impact from the COVID-shut down. The study has merit for change and insights that centers Indigenous youth voices but my main concern is in your methods, its delivery, language and coherency with the claims of citizen science.

Firstly, don’t you think the youth involved as co-researchers in your work should be cited as co-authors also? In the spirit of citizen science?

Abstract

Apart from the tautology in the description of your study design “qualitative, pre-post, quasi-experimental citizen science study involved eight Indigenous youth citizen scientists (YCS),” you conclusion is lacking in summarizing the essence of the study. Your conclusion should summarize key findings related to your study objective, provide practical implications and suggest future directions(if any). Rather you focus on something how the methodology demonstrates something unrelated to your primary study objective, which is not an answer to the original question set out to explore. Or is this a methods paper? Except you wanted to show how using an hybridized study design as you did, I suggest to stick to the original objective and other sections should logically flow out of that.

Introduction/methods

Well articulated and executed introduction starting the the problem theme from a global stage narrowing to children and youth. But nothing said (with regards to statistical figures and prevalences)about Indigenous youth and how it impacted them relative to non-Indigenous youth (given that your study wants to focus on Indigenous).

The abrupt jump from coping strategies to specialized mental health services makes readability difficult. If your aim is to emphasize the value of community-engaged research that was not a good work through to that point.

I also have some misgivings around the use of “Indigenous” as a broad-based terminology. Also, the claim they were citizen scientist (a term used by public health researchers to describe partners in a participatory research setting). Whie that is not a hill to die on, consider defining your use of the terminology.

Also per TCPS chapter 9 rules (which you cited), and the Truth and Reconciliation Commission calls to action, research involving Indigenous people in Canada must reflect their culture and values and co-created/driven by them in the spirit of self-determination and self-governance respecting their rights to data sovereignty. I fear I don’t see enough of these reflected in your work, how the relationships were built or how cultural safety was honored in including Elders and traditional knowledge holders in the processes of your community-engaged work. Granted your section overviewing the SIY initiative touched on them but you spend a huge portion is dedicated to describing/explaining the methodology (CS and TES) from a top-down rather than a bottom-up culturally grounded approach, makes it unrepresentative of true engagement or cultural humility/sensitivity. A section of your work should focus on the process of engagement rather than lumping them in between sections or using general statements like: “The team has built strong partnerships with participating Indigenous communities and adopted the Smart Framework…”

If community was engagement the language should reflect that is all I am saying. Who decided on the choice of study design, design of questions, carried out analysis, and interpretation, etc. state these for readers to appreciate the rigor of your relationship building process. The smart framework in my opinion read top down (designed by the research team) and forced on the youth participants rather than co-designed by and with them as it should in a true CS methodology.

For the sake of readers misconstruing your intent, it would be important to state how the youth were engaged from the beginning of this work. Often times we cite work involving Indigenous youth as youth-led or youth-engaged and the level of engagement is nothing short of extractive or helicopter research. If youth were truly engaged provide justification for why the team chose to carry interviews virtually (that’s not signifying true engagement, even though I understand it was around the time of COVID). I want more depth in explaining or justifying your claims is all I mean. I see the intent but I don’t know that others might the way you word some of the sections outlining engagement science.

6. PLOS authors have the option to publish the peer review history of their article (what does this mean? ). If published, this will include your full peer review and any attached files.

**Do you want your identity to be public for this peer review?** For information about this choice, including consent withdrawal, please see our Privacy Policy .

Reviewer #1: **Yes: ** Dr Nombuso Gama

Reviewer #2: **Yes: ** Udoka Okpalauwaekwe

---

## [Editor Report · Decision Letter 1]

29 Apr 2025

Mental Health Challenges and Resilience Strategies of Indigenous Youth Citizen Scientists Living in Rural Areas During COVID-19 School Closures

PMEN-D-24-00569R1

Dear Dr Katapally,

We are pleased to inform you that your manuscript 'Mental Health Challenges and Resilience Strategies of Indigenous Youth Citizen Scientists Living in Rural Areas During COVID-19 School Closures' has been provisionally accepted for publication in PLOS Mental Health.

Best regards,

Karli Montague-Cardoso

Executive Editor

PLOS Mental Health